# The Impact of Acute Ingestion of a Ketone Monoester Drink on LPS-Stimulated NLRP3 Activation in Humans with Obesity

**DOI:** 10.3390/nu12030854

**Published:** 2020-03-23

**Authors:** Helena Neudorf, Étienne Myette-Côté, Jonathan P. Little

**Affiliations:** School of Health and Exercise Sciences, University of British Columbia, Okanagan Campus, Kelowna, BC V1V 1V7, Canada; helena.neudorf@ubc.ca (H.N.); etmyco@gmail.com (É.M.-C.)

**Keywords:** ketones, obesity, immunometabolism, inflammation, caspase-1, inflammasome, beta-hydroxybutyrate, 3-hydroxybutyric Acid, interleukin-1β, NLR family, pyrin domain containing-3 protein

## Abstract

Activation of the NOD-like receptor pyrin-domain containing 3 (NLRP3) inflammasome is associated with chronic low-grade inflammation in metabolic diseases such as obesity. Mechanistic studies have shown that β-hydroxybutyrate (OHB) attenuates activation of NLRP3, but human data are limited. In a randomized, double-blind, placebo-controlled crossover trial (*n* = 11) we tested the hypothesis that acutely raising β-OHB by ingestion of exogenous ketones would attenuate NLRP3 activation in humans with obesity. Blood was sampled before and 30 min post-ingestion of a ketone monoester drink ((R)-3-hydroxybutyl (R)-3-hydroxybutyrate, 482 mg/kg body mass) or placebo. A 75 g oral glucose load was then ingested, and a third blood sample was obtained 60 min following glucose ingestion. NLRP3 activation was quantified by assessing monocyte caspase-1 activation and interleukin (IL)-1β secretion in ex vivo lipopolysaccharide (LPS)-stimulated whole-blood cultures. LPS-stimulated caspase-1 activation increased following glucose ingestion (main effect of time; *p* = 0.032), with no differences between conditions. IL-1β secretion did not differ between conditions but was lower 60 min post-glucose ingestion compared to the fasting baseline (main effect of time, *p* = 0.014). Plasma IL-1β was detectable in ~80% of samples and showed a decrease from fasting baseline to 60 min in the ketone condition only (condition × time interaction, *p* = 0.01). In individuals with obesity, an excursion into hyperglycemia following ingestion of a glucose load augments LPS-induced activation of caspase-1 in monocytes with no apparent impact of raising circulating β-OHB concentration via ingestion of exogenous ketones. Exogenous ketone supplementation may impact plasma IL-1β, but these findings require confirmation in studies with larger sample sizes.

## 1. Introduction

Innate immune dysfunction is implicated in a variety of chronic conditions, including obesity-related metabolic diseases [1,2,3]. This dysfunction is thought to be mediated in part by the NOD-like receptor pyrin-domain containing 3 (NLRP3) inflammasome, a pattern recognition receptor. The ability of NLRP3 to mount a response to pathogen-associated molecular patterns (PAMPs) is usually a crucial protective mechanism. However, NLRP3 also responds to endogenously produced damage-associated molecular patterns (DAMPs) [4]. These can be products of tissue damage or byproducts of fuel metabolism, such as hyperglycemia [2]. 

Canonical NLRP3 signaling occurs in the presence of two stimuli: first, a priming stimulus, such as lipopolysaccharide (LPS) binding to toll-like receptor (TLR)-4, which functions to upregulate the transcription of NLRP3 and pro-interleukin (IL)-1β via nuclear factor κ-light-chain-enhancer of activated B cells (NF-κB) signaling. A secondary stimulus is then required to activate NLRP3. The second activating stimulus causes assembly of NLRP3 by recruitment of apoptosis-associated speck-like protein containing a caspase activation and recruitment domain (CARD) (ASC) and pro-caspase-1, which auto-activates into its active form. In human leukocytes, the NLRP3 inflammasome can also be activated by a single stimulus, such as LPS [5]. Both types of signaling converge on the activation of caspase-1, which cleaves pro-IL-1β and pro-IL-18 into their mature forms to be secreted from the cell [2]. The resulting secretion of cytokines contributes to the perpetuation of a chronic inflammatory cycle [2]. 

Overactivation of the NLRP3 inflammasome contributes to accelerated pathogenesis and complications of obesity-related conditions such as type 2 diabetes (T2D), atherosclerosis, and gout [1,2,3]. For example, Maedler and colleagues found that pancreatic beta-cells cultured in high glucose upregulate IL-1β secretion [6]. Exposure of human monocytes to high glucose causes activation of the NLRP3 inflammasome and promotes release of IL-1β and IL-18 [7,8]. Furthermore, activation of NLRP3 has been shown to be critical to the development of obesity-induced insulin resistance in mice and humans [9].

Ketone bodies, with their ability to act as immunomodulatory signaling metabolites [6,7,8,9], hold potential to slow progression of obesity-related diseases. Specifically, the ketone body β-hydroxybutyrate (β-OHB) has been shown to block activation of pro-caspase-1 into caspase-1, and consequently attenuate formation of mature IL-1β in stimulated mouse and human leukocytes [10,11,12] in an NLRP3 inflammasome-specific manner [13]. 

To our knowledge, we are the only group thus far to investigate the effect of elevated β-OHB on NLRP3 activation in humans [14]. Using both ketone salts and a ketone monoester drink (KMD), we have previously shown that acute elevation of blood β-OHB concentration in young, healthy humans augmented LPS-induced caspase-1 activation in blood leukocytes [14]. However, basal levels of inflammation are low in this population, and it is unknown how raising β-OHB with exogenous ketone ingestion affects NLRP3 activation in individuals who are at risk for developing metabolic diseases (e.g., obesity). Therefore, the primary purpose of this study was to investigate the effect of exogenous ketone ingestion on markers of NLRP3 activation in individuals with obesity. A secondary purpose was to examine how prior ingestion of a KMD impacted inflammatory responses to acute hyperglycemia. We tested the central hypothesis that markers of NLRP3 activation and inflammation would be attenuated in response to raising circulating β-OHB via ingestion of a KMD.

## 2. Materials and Methods 

### 2.1. Ethical Approval

Ethical approval for this study was obtained from the University of British Columbia Clinical Research Ethics Board (ID H16-01846). The study was conducted in accordance with the Declaration of Helsinki. The study was registered as a clinical trial on clinicaltrials.gov (NCT03461068) and the current investigation represents a sub-study reporting the pre-registered secondary outcomes related to inflammation. Written informed consent was obtained from all participants prior to beginning data collection.

### 2.2. Participants

This study was conducted as part of a larger study [15] for which 15 participants (5 males, 10 females) were recruited. Due to technical and scheduling considerations, full data for inflammatory measures relevant to this paper were obtained from 12 participants (4 males, 8 females). One participant was excluded to meet statistical assumptions (see Section 2.8) such that data from 11 participants (4 males, 7 females) were included in the analysis. Participants were included if they were 30–65 years of age, had a waist circumference in the obesity range (greater than 102 cm for males and 88 cm for females), as well as a body mass index (BMI) ≥28 kg/m^2^ (high overweight category). This was done in order to avoid inclusion of participants with an elevated waist circumference and lean BMI. Prospective participants were excluded if they were taking medications known to affect glucose or lipid metabolism, following an intensive exercise regime (defined as following a training program consisting of >300 min per week of physical activity), diagnosed with diabetes or heart disease, were pregnant, or had an acute infection at the time of data collection. Participants were also excluded if they were following a very-low-carbohydrate diet or intermittent fasting protocol, so as to exclude individuals who might have elevated fasting ketone levels at baseline. This was assessed by self-report and confirmed by 24 h dietary recall along with measurement of fasting ketone levels prior to each condition. A graduate student research assistant was trained by a registered dietician to collect these measures. Participant characteristics are reported in Table 1.

### 2.3. Experimental Procedure

The present study followed a placebo-controlled, double-blind, randomized crossover design. There were three total visits to the laboratory for each participant: the first was to conduct screening, obtain written informed consent, and familiarize the participant with the laboratory. A menstrual cycle questionnaire was also completed during this time, as testing was only conducted during the follicular phase (days 3–9) in women that were premenopausal. The second and third visits involved participation in each of the experimental conditions. Participants reported to the laboratory after fasting for at least 10 h and having refrained from alcohol and exercise for the previous 24 h. Compliance was assessed by self-report. On the day prior to the second visit, participants completed a 24 h food log and were asked to replicate this menu on the day prior to their third visit. Participants were provided with a copy of their food log after the second visit and compliance was confirmed by a researcher upon arrival at the third visit. The washout period between conditions was a minimum of 48 h.

A fasted baseline blood sample (−30 min) was drawn from the antecubital vein through an indwelling intravenous catheter (BD Nexiva, Utah, USA) into an EDTA vacutainer (BD Vacutainer, Utah, USA). The KMD (ΔG® from TΔS® Ltd., UK, 0.45 mL/kg body weight corresponding to 482 mg/kg body weight) or isovolumetric placebo was then consumed. Following 30 min of quiet sitting, a second blood sample was collected (0 min). Participants then consumed a 75 g oral glucose tolerance test (OGTT) drink (Thermo Fisher Scientific, Waltham, MA, USA). Time points were named relative to the consumption of the OGTT drink. Following a further 60 min of quiet sitting, a third blood draw was obtained (60 min). This protocol allowed for testing the immediate impact of raising blood β-OHB (−30 vs. 0 min) as well as for determining how prior ingestion of the KMD might impact the expected hyperglycemia-induced activation of the NLRP3 pathway (−30 vs. 60 min). 

### 2.4. Ketone Monoester Drink

The KMD (ΔG® from TΔS® Ltd., UK) was provided as a clear liquid mixed with water and a flavored sugar-free sweetener (Mio, Kraft Foods). After ingestion, the ketone monoester compound, (R)-3-hydroxybutyl (R)-3-hydroxybutyrate, is cleaved by non-specific carboxylesterases located in tissues such as the gastrointestinal tract, blood, and liver to yield β-OHB and 1,3-butanediol, the latter being further metabolized to β-OHB in the liver. Individual dosing of the KMD was calculated as 0.45 mL/kg body weight, the equivalent of 482 mg (R)-3-hydroxybutyl (R)-3-hydroxybutyrate per kg of bodyweight. This dosing strategy has been shown to increase circulating β-OHB concentration to ~3 mM within 15–30 min [14]. The placebo was volume matched and consisted of water mixed with the same flavored sugar-free sweetener and a bitter agent (Bitrex, Scotland, UK) to mimic the taste of the KMD. Participants wore a nose clip during ingestion of the drinks and were given 20 mL of calorie-free sports drink (Gatorade G2, Chicago, IL, USA) immediately after ingesting the ketone monoester or placebo drink to remove any lingering flavor. 

### 2.5. Quantification of Active Caspase-1

Presence of activated caspase-1 was quantified by flow cytometry (MACSQuant Analyzer 8, Miltenyi Biotec, Bergisch Gladbach, Germany) using the FAM-FLICA® Caspase-1 Assay Kit (ImmunoChemistry Technologies, LLC, Bloomington, MN, USA) as previously described [14]. Briefly, whole blood was aliquoted and stimulated with either LPS (10 ng/mL final concentration in culture) or left unstimulated (control) for 60 min at 37 °C and 5% CO_2_ in the dark. The fluorescent inhibitor probe FAM-FLICA® (consisting of FAM-YVAD-FMK) was added to the culture to stain active caspase-1, and cultures were incubated for a further 50 min. To identify monocytes, cluster of differentiation (CD)14 VioBlue (Miltenyi Biotech) was then added during the final 10 min of incubation. After a total of 120 min of incubation, red blood cells were lysed, and two washes were conducted to remove red blood cell debris. Propidium iodide was added to stain dead cells, and samples were immediately analyzed by flow cytometry. A hierarchical gating strategy was used to acquire data. Flow cytometry data were analyzed using MACSQuantify^TM^ (Version 2.6, Miltenyi Biotec, Bergisch Gladbach, Germany). 

### 2.6. Quantification of IL-1β Secretion

Whole blood from each time point was aliquoted and stimulated with LPS (10 ng/mL final concentration in culture) or left unstimulated (control). Following 120 min of incubation at 37 °C in 5% CO_2_, supernatants were collected and frozen at −80 °C for later analysis. Supernatants were batch analyzed in duplicate with the Mesoscale Discovery (MSD) T-PLEX Human Pro-Inflammatory Panel I following the manufacturers’ instructions, and plates were read on an MSD QuickPlex SQ 120 Reader. This assay measures IL-1β, IL-6, tumor necrosis factor (TNF-α), and interferon (IFN)-γ. The lower limit of detection (LLOD) for IL-1β was 0.3 pg/mL, IL-6 was 0.19 pg/mL, TNF-α was 0.56 pg/mL, and IFN-γ was 2.2 pg/mL. Only two full data sets (out of 12) were above the detection limit for IFN-γ, therefore it is not reported.

### 2.7. Quantification of Plasma Cytokines

EDTA blood from each timepoint was centrifuged at 1550 g for 15 min at 4 °C. The plasma was then aliquoted and frozen at − 80°C for later analysis. Frozen plasma samples were thawed and batch analyzed in duplicate with the Mesoscale Discovery V-PLEX Proinflammatory Panel 1 (Human IL-1β, IL-6, TNF-α), Mesoscale Discovery V-PLEX Human Vascular Injury Panel (Human C-reactive protein (CRP)), and Mesoscale Discovery U-PLEX Human IL-18 Assay kits. IL-1β was undetectable in ~21% of samples (14/66 total blood samples) resulting in complete data for both conditions in 4 participants and partial data for the other 7 participants for this cytokine. The LLOD for hs-CRP was 1.33 pg/mL, IL-18 was 2.5 pg/mL, IL-1β was 0.05 pg/mL, Il-6 was 0.06 pg/mL, and TNF-α was 0.04 pg/mL.

### 2.8. Statistical Analysis

All statistical analyses were performed using RStudio (Version 3.5.1 “Feather Spray”, Vienna, Austria) [16]. Presence of influential data points was assessed using Cook’s distance using the influence.ME package in R [17]. A data point was considered influential if Cook’s distance was above the cutoff, which was calculated as 4/N. One participant’s data met this criteria at multiple timepoints and was therefore excluded from the analysis resulting in *n* = 11. Additionally, one data set from plasma IL-1β in the ketone condition was also determined to be influential at multiple timepoints and was therefore excluded from the analysis resulting in *n* = 10 in the ketone condition and *n* = 11 in the placebo condition for this outcome measure. Homoscedasticity was assessed visually by plotting residuals vs. fitted values. Normality and linearity of data were assessed by plotting a normal Q–Q plot and a histogram of residuals. IL-1β data from 2 h cultures required log transformation to fit these assumptions. 

Analysis was performed by use of a linear mixed effects model using the lme4 package in RStudio [18]. The model was built with fixed effects of time (-30, 0, and 60 min) and condition (placebo vs. ketone) using an interaction term in the model. The random effect included was a random intercept for each participant [18]. To test for an interaction effect, a full model was compared against a reduced model using a likelihood ratio test. In the absence of an interaction effect, this was followed up with comparison against a null model in which either fixed effect was removed to test for the presence of a main effect. Significant interactions were followed up using within-condition pairwise comparisons using a Bonferroni correction and main effects of time were followed up with pairwise comparisons for all timepoints with conditions collapsed. Significance was assessed at *P* < 0.05 and 95% confidence intervals are reported (CI_.95_). Cohen’s *d* effect sizes were calculated for pairwise comparisons with a Hedge’s *g* correction for small sample size using the Effsize package [19]. Missing data for IL-1β were not imputed as linear mixed models can handle missing data. 

A sample size calculation was conducted to accommodate the primary outcome of glucose area under the curve for the main research question reported in Myette-Côté et al. [15]. Consequently, given the exploratory nature of the outcomes relevant to the present investigation, a formal sample size calculation was not conducted for active caspase-1 or cytokine secretion. We were not powered to conduct any statistical analysis to look for sex differences in this study but have included Appendix A with data disaggregated by sex as Appendix A.

## 3. Results

### 3.1. Acute Ingestion of KMD Effectively Raises Blood β-OHB Levels

A significant condition × time interaction was found for blood β-OHB (*p* < 0.001, *n* = 11, Figure 1). Post hoc testing found that following ingestion of the KMD, blood β-OHB increased from 0.21 ± 0.07 mM at −30 min to 1.99 ± 0.67 mM at 0 min (*p* < 0.001, CI_.95_ (2.3, 1.3), *g* = 3.3), and then to 2.96 ± 0.91 mM at 60 min (*p* < 0.001, CI_.95_ (1.3, 0.6), *g* = 1.2). Blood β-OHB did not change following ingestion of the placebo between −30 and 0 min (*p* > 0.999, CI_.95_ (−0.03, 0.05), *g* = 0.1), or 0 and 60 min (*p* > 0.999, CI_.95_ (−0.04, 0.03), *g* < 0.01).

### 3.2. No Impact of KMD on Caspase-1 Activation in Unstimulated Whole-Blood Cultures

Although basal caspase-1 is low in unstimulated cells, it was of interest to determine whether the KMD altered basal caspase-1 activation in unstimulated whole-blood cultures. However, there was no difference in basal caspase-1 activation between the placebo and ketone monoester drink (Figure 2B).

### 3.3. KMD Does Not Attenuate Caspase-1 Activation in LPS-Stimulated Monocytes

Given the low levels of NLRP3 activation in unstimulated cells, whole-blood cultures were stimulated with LPS, a known activator of NLRP3 in human monocytes. A significant main effect of time was found for LPS-stimulated caspase-1 activation (*p* = 0.043, *n* = 11, Figure 2C). Post hoc pairwise comparisons (conditions collapsed) found that LPS-stimulated caspase-1 activation in monocytes did not change significantly from −30 to 0 min (*p* = 0.420, CI_.95_ (−0.1, 0.3), *g* = 0.3) or from 0 to 60 min (*p* = 0.147, CI_.95_ (0.001, 0.4), *g* = 0.3). However, caspase-1 activation was ~14% higher at 60 min when compared to −30 min (*p* = 0.013, CI_.95_ (−0.5, −0.1), *g* = 0.5). 

### 3.4. KMD Does Not Impact Secretion of IL-1β, IL-6, or TNF-α in LPS-Stimulated Whole-Blood Cultures

#### 3.4.1. IL-1β

IL-1β secretion from LPS-stimulated whole-blood cultures demonstrated a main effect of time (*p* = 0.014, *n* = 11, Figure 3A). IL-1β secretion was 12.5 ± 11.0 pg/mL at −30 min, 13.1 ± 8.0 pg/mL at 0 min, and 8.8 ± 8.1 pg/mL at 60 min in the ketone monoester condition. In the placebo condition, IL-1β secretion was 12.4 ± 8.1 pg/mL at −30 min, 15.2 ± 13.3 p/mL at 0 min, and 9.8 ± 9.2 pg/mL at 60 min. Post hoc pairwise comparisons (conditions collapsed) found no difference between −30 and 0 min (*p* = 0.170, CI_.95_ (−0.1, 5.9), *g* = 0.3) or between −30 and 60 min (*p* = 0.428, CI_.95_ (−0.8, 5.7), *g* = 0.3). However, IL-1β secretion at 60 min was significantly reduced from that at 0 min (*p* = 0.014, CI_.95_ (1.6, 8.0), *g* = 0.4).

#### 3.4.2. IL-6

IL-6 secretion from LPS-stimulated whole-blood cultures did not change over time or in response to the KMD (Figure 3B). In the ketone condition, IL-6 secretion was 38.8 ± 29.4 pg/mL at −30 min, 45.4 ± 32.2 pg/mL at 0 min, and 34.3 ± 24.5 pg/mL at 60 min. In the placebo condition, IL-6 secretion was 42.8 ± 26.5 pg/mL at −30 min, 43.1 ± 27.4 pg/mL at 0 min, and 39.7 ± 30.3 pg/mL at 60 min.

#### 3.4.3. TNFα

A main effect of time was found for TNFα secretion from LPS-stimulated whole-blood cultures (*p* = 0.009, *n* = 11, Figure 3C). TNFα secretion was 140 ± 88.8 pg/mL at −30 min, 169.2 ± 107.6 pg/mL at 0 min, and 104.0 ± 54.4 pg/mL at 60 min in the ketone monoester condition. In the placebo condition, secreted TNFα was 158.0 ± 72.0 pg/mL at −30 min, 167.4 ± 93.4 pg/mL at 0 min, and 141.4 ± 77.5 pg/mL at 60 min. Post hoc pairwise comparisons (conditions collapsed) found no difference between −30 and 0 min (*p* = 0.320, CI_.95_ (−5.3, 50.3), *g* = 0.2) or between −30 and 60 min (*p* = 0.190, CI_.95_ (−56.1, 1.7), *g* = 0.2). However, TNFα secretion was significantly reduced at 60 min compared to that at 0 min (*p* = 0.009, CI_.95_ (17.7, 73.8), *g* = 0.5).

### 3.5. Plasma Cyotkines

In order to better characterize participants’ baseline and in vivo responses, plasma cytokines were measured by MSD assays (Table 2). 

#### 3.5.1. C-Reactive Protein

A main effect of condition was found for hs-CRP (*p* = 0.027, *n* = 11, Table 2, and Appendix A). However, post hoc pairwise comparisons found no difference between conditions at −30 min (*p* = 0.388; CI_.95_ (−0.29, 0.07); *g* = 0.4), 0 min (*p* = 0.773; CI_.95_ (−0.31, 0.13); *g* = 0.2), or 60 min (*p* > 0.999; CI_.95_ (−0.23, 0.14); *g* = 0.1).

#### 3.5.2. Plasma Interleukin-1β

A time × condition interaction was found for IL-1β (*p* = 0.01, *n* = 11, Table 2, and Appendix A). Post hoc pairwise comparisons in the ketone condition found no difference between −30 and 0 min (*p* = 0.194; CI_.95_ (−0.002, 0.06); *g =* 0.1) or 0 and 60 min (*p* = 0.254; CI_.95_ (−0.01, 0.13); *g =* 0.5). However, IL-1β at 60 min was significantly reduced from baseline −30 min in the ketone condition (*p* = 0.002; CI_.95_ (0.05, 0.10); *g =* 0.98). In the placebo condition, there were no differences between any time points (−30 vs. 0 min: *p* > 0.999, CI_.95_ (−0.05, 0.06); *g =* 0.3; 0 vs. 60 min: *p* = 0.696, CI_.95_ (−0.04, 0.02); g = 0.3; −30 vs. 60 min: *p* > 0.999, CI_.95_ (−0.04, 0.02), *g* = 0.03).

#### 3.5.3. Plasma Interleukin-18

A main effect of time was found for IL-18 (*p* = 0.039, *n* = 11, Table 2, and Appendix A). Post hoc pairwise comparisons (conditions collapsed) found that plasma IL-18 was significantly reduced from −30 to 0 min (*p* = 0.011; CI_.95_ (33.90, 151.23); *g* = 0.3). However, there was no difference between −30 and 60 min (*p* = 0.271; CI_.95_ (−15.22, 193.55); *g* = 0.3) or between 0 and 60 min (*p* > 0.999; CI_.95_ (−87.64, 80.33); *g* = 0.2).

#### 3.5.4. Plasma Interleukin-6

No interaction or main effects were found for IL-6 (time × condition interaction: *p* = 0.109, main effect of time: *p* = 0.082, main effect of condition: *p* = 0.435, Table 2, and Appendix A).

#### 3.5.5. Plasma Tumor Necrosis Factor-α

A main effect of time (*p* = 0.018) and a main effect of condition were found (*p* = 0.031, *n* = 11, Table 2, and Appendix A). Post hoc pairwise comparisons (conditions collapsed) found that plasma TNFα was significantly reduced from −30 to 0 min (*p* = 0.033; CI_.95_ (0.03, 0.23); *g* = 0.1). There were no differences between −30 and 60 min (*p* = 0.128; CI_.95_ (0.006, 0.31); *g* = 0.1) or 0 and 60 min (*p* > 0.999; CI_.95_ (−0.14, 0.11); *g* = 0.1).

## 4. Discussion

The major finding of this study was that elevation of blood β-OHB by ingestion of a KMD did not alter LPS-stimulated caspase-1 activation or IL-1β, IL-6, or TNFα secretion measured in whole-blood cultures in the basal state or during an excursion into hyperglycemia, although ingestion of 75 g of glucose appeared to have independent effects of increasing caspase-1 activation and altering cytokine secretion. The finding that elevated β-OHB did not inhibit NLRP3 inflammasome activation is contrary to the existing animal and in vitro literature [11,12,13]. In addition, the results appear at odds with our previous investigation showing that acute ingestion of a ketone salt or KMD augmented LPS-induced caspase-1 activation in whole-blood cultures in young, healthy individuals [14].

Elevated NLRP3 signaling activity is associated with obesity and obesity-related chronic diseases [2,9,20]. In the present study, we expected that individuals with obesity would be more permissive to detecting the potential effects of β-OHB to inhibit NLRP3 activation due to the propensity for elevated baseline inflammation (as evidenced by mean fasting plasma hs-CRP concentration of ~0.4 mg/dL in our sample). However, it appears that acutely increasing β-OHB via ingestion of a KMD does not impact activation of the NLRP3 pathway, as assessed by caspase-1 activation and IL-1β secretion in LPS-stimulated whole-blood cultures. One possibility is that raising β-OHB through KMD does not inhibit the NLRP3 pathway in circulating human monocytes or that longer exposure to elevated β-OHB concentration is needed to see inhibitory effects. We did, however, detect a significant decrease in plasma IL-1β at 60 min in the ketone condition but not the placebo (condition × time interaction, Table 2). These findings are intriguing but should be interpreted with caution as the analyses of plasma cytokines was exploratory in nature, and due to undetectable values at several timepoints (which is typical for plasma IL-1β analyses) the data are limited to only ~80% of the blood samples with only 4 participants having detectable values for all timepoints in both conditions. 

The work by Youm and colleagues demonstrated that β-OHB was shown to be an effective NLRP3 inhibitor when human monocytes were stimulated with 1000 ng/mL LPS, and anti-inflammatory effects were seen primarily at 10 and 20 mM β-OHB [13]. In whole-blood cultures, we stimulated with 10 ng/mL LPS (which gives maximal stimulation of cytokines in our experimental setup). In vivo β-OHB levels from ingestion of the KMD were ~3 mM. This is similar to the levels achieved in mild ketosis from ~3 days of fasting or ketogenic diet (1–3 mM) but less than the levels achieved during a fast exceeding one week (5–8 mM) [21,22,23]. In contrast, the levels achieved during diabetic ketoacidosis may rise as high as 25 mM [22,23], which is more similar to the concentration at which the maximal anti-inflammatory effect of β-OHB has been shown in cell culture [13]. If higher concentrations of LPS and β-OHB are needed to uncover anti-inflammatory effects, it would seem unlikely that exogenous ketones could attenuate physiologic levels of inflammation given the limits of increasing blood β-OHB with acute KMD ingestion and that circulating LPS concentrations are orders of magnitude lower in physiologically relevant conditions of sepsis and metabolic endotoxemia [24]. 

Interestingly, Thomsen and colleagues found higher circulating IL-1β concentration in response to in vivo LPS stimulation alone when β-OHB was elevated by infusion compared to lipid or placebo infusion in healthy humans [25]. This supports the idea that elevated β-OHB may not be universally “anti-inflammatory” in humans. This is also supportive of the NLRP3 inflammasome in human models only requiring a single stimulus from LPS alone, rather than the dual stimulus required in murine models [5]. The manner in which the human NLRP3 inflammasome is activated (single versus dual stimulus) may be important: Youm and colleagues showed that the mechanism by which β-OHB is able to inhibit the NLRP3 inflammasome is by inhibition of K^+^ efflux from the cell [13]. Gaidt and colleagues showed that when the human NLRP3 inflammasome is activated by LPS alone, the activating signal is not transmitted via the canonical pathway [5]. Rather, the NLRP3 inflammasome is activated via an alternative pathway that signals through the toll-like receptor (TLR)-adaptor TIR-domain-containing adapter-inducing interferon-β (TRIF), a molecule that is not involved in canonical NLRP3 signaling [5]. Importantly, this alternative signaling cascade occurs independent of K^+^ efflux [5]. In further support of this, β-OHB does not attenuate activation of murine caspase-11, which is activated by LPS–TLR4–TRIF signaling [13]. Taken together, these results provide a likely reason why β-OHB did not inhibit the NLRP3 inflammasome pathway in our single LPS stimulus model.

Exposure time to β-OHB may also be a crucial component in order to see any beneficial effects. In vivo exposure time in our study was ~30 and 90 min at each timepoint following baseline. In vivo murine models have used an exposure period of 24 h with β-OHB delivered by pump treatment [26] or treatment with β-OHB nanolipogels [13], or five days of either β-OHB injection [10] or ketogenic diet [11]. In vitro models have used an exposure period of as little as 60 min [11,13] to as much as 24 h [11,26]. In order to fully explore the potential impact of β-OHB on NLRP3 inflammasome activation in humans, studies using multiple doses of KMD may be needed so that cells are exposed to elevated β-OHB for more prolonged periods of time. We are currently working on such studies [Clinical Trials Identifier: NCT03817749]. 

Another interesting observation from this study was finding that LPS-stimulated IL-1β and TNFα secretion was reduced at 60 min compared to that at 0 min (Figure 3A,C). Because IL-1β is activated downstream of caspase-1, it was expected that the IL-1β response would mirror that of caspase-1 activation, but this did not appear to be the case. LPS-induced caspase-1 activation increased at 60 min following glucose ingestion at the timepoint when IL-1β and TNFα secretion was reduced compared to baseline. One possible explanation may be that secretion of mature IL-1β depends not only on caspase-1 activation but also on the presence of pro-IL1β. Previously, our group showed no change in NLRP3 or *IL1B* mRNA following ingestion of a KMD [14], therefore it is possible that inadequate levels of pro-IL1β were present to secrete at 60 min. However, this remains speculative as we did not measure *IL1B* transcription or intracellular pro-IL1β protein in this study. 

### Limitations and Future Directions

While we have made an effort in the present study to translate previous in vitro and animal work into a clinically relevant human model, our ex vivo whole-blood culture methods still may not properly assess the clinical relevance of elevated β-OHB on inflammation. Here, we have attempted to raise blood β-OHB to physiologic levels to create a relevant model of ketone exposure; however, testing the impact on innate immune activity with ex vivo stimulation may not explicitly translate to modulation of inflammation in a clinical condition. Additionally, due to the relatively abrupt elevation in blood ketones achieved by consuming the KMD, the findings may not be directly applicable to dietary modifications that induce endogenous ketogenesis more gradually. However, this trade-off was necessary in order to assess the independent effects of the β-OHB on innate immune signaling pathways such as the NLRP3 inflammasome. When produced endogenously, β-OHB is a molecule that coincides with fasting/starvation, and thus likely interacts with many signaling cascades related to such a physiological state.

The complexity of in vivo inflammation is also reflected by the wide variety of molecules which may activate the NLRP3 inflammasome through various signaling pathways. Because we only tested our hypothesis using a single NLRP3 activator, it remains to be determined whether raising blood β-OHB through exogenous ketones will affect NLRP3 in response to other stimuli. As such, our findings may only be applicable to models in which NLRP3 is activated by LPS. Metabolic endotoxemia, in which LPS enters the circulation from the digestive tract, is thought to contribute to the development of obesity, insulin resistance, and cardiovascular disease [24], which is why we chose this stimulus. However, other PAMPs and DAMPs (e.g., reactive oxygen species, cholesterol crystals, fatty acids) are likely also involved in perpetuating chronic inflammation in obesity-related metabolic diseases [2,27]. Therefore, future research should investigate the ability of β-OHB to directly inhibit the NLRP3 inflammasome in response to other relevant stimuli, perhaps reflecting classical (canonical) signaling versus alternative (non-canonical) signaling. Additionally, our results suggest that the increase in caspase-1 activation and reduction in IL-1β and TNFα secretion were related to the increase in glycemia from the OGTT and not impacted by differences in blood β-OHB concentrations. Although the concentration of plasma glucose at 60 min was not statistically different in the present analyses (data not shown), 2 h glucose area under the curve was significantly reduced in response to elevated blood β-OHB (all glucose data are reported in Myette-Côté et al. [15]). Future work is needed to elucidate potential interactions between glucose, β-OHB, and NLRP3 activation in humans. Finally, while our pilot study was small, it was strengthened by the inclusion of both males and females in the sample. However, due to the small sample size, we were unable to explore any potential sex-specific responses to exogenously elevated ketones. Future research should aim to specifically address this question.

## 5. Conclusions

Our findings demonstrate that in there is no apparent impact of acutely raising blood β-OHB concentration via KMD ingestion on activation of the NLRP3 inflammasome measured in LPS-stimulated whole-blood cultures. It may be that acutely elevating β-OHB with exogenous ketones does not inhibit the NLRP3 inflammasome in the same manner as high β-OHB concentrations appear to do in cell culture and animal models [10,11,12,13], or that longer exposure times or different inflammatory stimuli are needed to uncover potential immunomodulatory effects of β-OHB in humans. Further research is needed to help determine how ketosis impacts inflammation and immune function in humans; exogenous ketone supplements may be a useful tool in these efforts.

## Figures and Tables

**Figure 1 nutrients-12-00854-f001:**
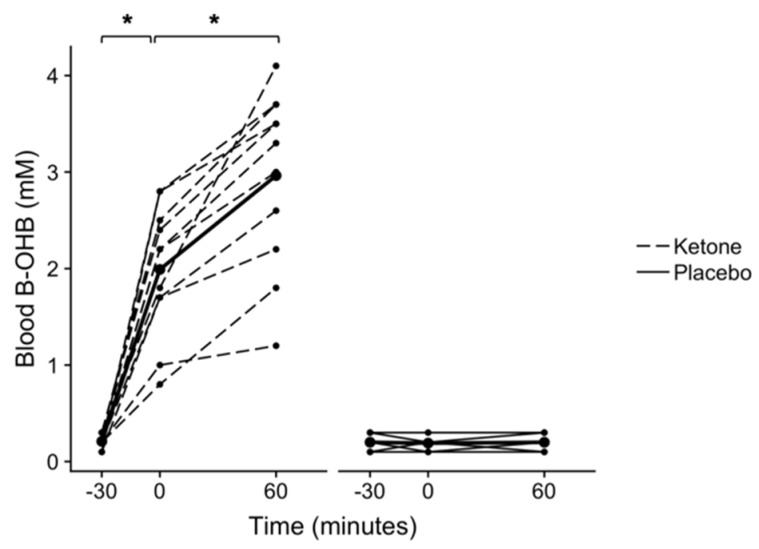
Blood beta-hydroxybutyrate (β-OHB) following ketone or placebo drink ingestion. Each drink was ingested immediately following the fasted −30 min blood draw. All timepoints are named relative to the “0 min” time point, at which the OGTT drink was consumed (i.e. −30 min occurred 30 minutes prior to the consumption of the OGTT drink). A time × condition interaction was found (*p* < 0.001). **p* < 0.001 Bonferroni-corrected within-condition pairwise comparisons. Individual participant data are shown with the ketone condition represented by dashed lines and the placebo condition represented by solid lines. Means are shown by bold lines with triangles. *n* = 11.

**Figure 2 nutrients-12-00854-f002:**
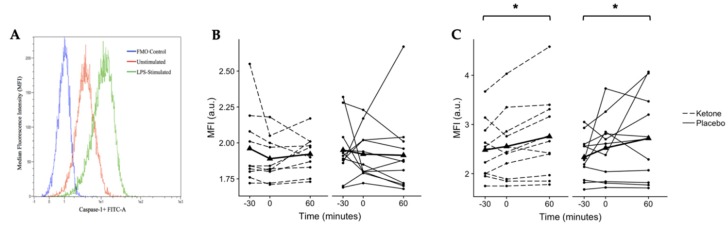
Median fluorescence intensity (MFI) representative of active caspase-1 in human monocytes as quantified by flow cytometry. (**A**) Histogram comparing MFI of a fluorescence minus one (FMO) control, an unstimulated sample, and an lipopolysaccharide (LPS)-stimulated sample. (**B**) No changes were observed in unstimulated human monocytes. (**C**) A significant main effect of time was found (*p* = 0.043) for LPS-stimulated human monocytes with post hoc testing revealing significantly higher caspase-1 activation at 60 vs. −30 min (**p* = 0.013, Bonferroni-corrected pairwise comparison). Individual participant data are shown with the ketone condition represented by dashed lines and the placebo condition represented by solid lines. Means are shown by bold lines with triangles. *n* = 11.

**Figure 3 nutrients-12-00854-f003:**
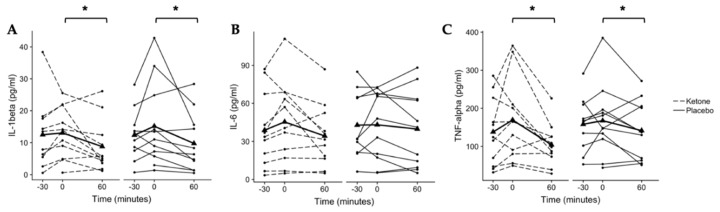
Secreted cytokines from LPS-stimulated whole-blood cultures. (**A**) A main effect of time was found for IL-1β secretion (*p* = 0.014) with a significant difference detected between 0 and 60 min (**p* = 0.014, Bonferroni-corrected pairwise comparison). (**B**) No changes were observed for interleukin (IL)-6 (*p* > 0.05). (**C**) A main effect of time was found for tumor necrosis factor (TNF)-alpha secretion (*p* = 0.009) with a significant difference detected between 0 and 60 min (**p* = 0.009 Bonferroni-corrected post hoc). Individual participant data are shown with the ketone condition represented by dashed lines and the placebo condition represented by solid lines. Means are shown by bold lines with triangles. *n* = 11.

**Table 1 nutrients-12-00854-t001:** Participant characteristics.

	Males (*n* = 4)	Females (*n* = 7)	All (*n* = 11)
Age (years)	45.0 ± 15.0	49.1 ± 9.2	47.6 ± 11.1
Body Mass (kg)	110.8 ± 15.0	83.6 ± 19.8	93.5 ± 22.2
Body Mass Index (kg/m^2^)	34.2 ± 4.2	32.5 ± 5.8	33.1 ± 5.1
Waist Circumference (cm)	108.9 ± 9.0	99.6 ± 17.6	103.3 ± 14.9
Systolic Blood Pressure (mmHg)	133.3 ± 7.2	121.6 ± 11.0	125.8 ± 11.1
Diastolic Blood Pressure (mmHg)	82.5 ± 9.3	83.4 ± 8.1	83.5 ± 8.1
Hemoglobin A1c (%)	5.4 ± 0.3	5.8 ± 0.7	5.7 ± 0.6
Fasting Glucose (mM)	5.4 ± 0.5	6.0 ± 1.1	5.8 ± 1.0
Fasting Insulin (pmol/L)	203.3 ± 90.0	144.9 ± 67.9	166.1 ± 80.2
OGTT Glucose (120 min, mM)	5.2 ± 2.1	7.1 ± 2.7	6.5 ± 2.6
OGTT Insulin (120 min, pmol/L)	778.0 ± 707.3	859.6 ± 736.1	827.0 ± 685.3

OGTT, oral glucose tolerance test. Data are presented as means ± standard deviation (SD).

**Table 2 nutrients-12-00854-t002:** Plasma cytokines.

	Ketone			Placebo		
	−30 Min	0 Min	60 Min	−30 Min	0 Min	60 Minutes
hs-CRP (mg/dL) *	0.39 ± 0.25	0.34 ± 0.25	0.34 ± 0.23	0.50 ± 0.28	0.43 ± 0.29	0.38 ± 0.28
IL-1β (pg/mL) ^‡^	0.11 ± 0.07	0.10 ± 0.12	0.05 ± 0.05^b^	0.05 ± 0.03	0.04 ± 0.03	0.05 ± 0.03
IL-18 (pg/mL) ^†^	2204.2 ± 831.5	2072.4 ± 779.7^a^	2147.9 ± 781.1	2141.0 ± 788.4	2087.7 ± 773.8^a^	2019.0 ± 722.1
Il-6 (pg/mL)	1.15 ± 0.41	1.21 ± 0.41	1.34 ± 0.35	1.11 ± 0.52	1.21 ± 0.46	1.24 ± 0.63
TNFα (pg/mL) *^,†^	3.61 ± 1.00	3.43 ± 0.80^a^	3.49 ± 0.84	3.51 ± 0.94	3.25 ± 0.85^a^	3.32 ± 0.79

Data are means ± standard deviation (SD). High sensitivity C-reactive protein (hs-CRP); interleukin (IL)-1β; interleukin (IL)-18; interleukin (IL)-6; tumor necrosis factor (TNF)-α. * Main effect of condition *p* < 0.05. ^†^ Main effect of time *p* < 0.05; ^‡^ Time by condition interaction *p* < 0.05. ^a^ Significantly different from −30 min, conditions collapsed (*p* < 0.05; Bonferroni-corrected post-hoc from main effect of time). ^b^ Significantly different from −30 min within-condition *p* < 0.05 (Bonferroni-corrected post-hoc). *n* = 11 for all (14/66 samples undetectable for IL-1β).

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
