# Peer review of "The Impact of Acute Ingestion of a Ketone Monoester Drink on LPS-Stimulated NLRP3 Activation in Humans with Obesity"

_nutrients, 2020, doi:10.3390/nu12030854_

Round 1

Reviewer 1 Report

Overview and Summary

The authors tested whether elevated ketones impact monocyte LPS-stimulated NLRP3 activation with a hyperglycemic excursion. They utilized a ketone monoester beverage to acutely raise circulating ketones followed by an OGTT to induce a hyperglycemic excursion in 11 individuals with obesity in a randomized, double-blind placebo controlled fashion. Monocytes were isolated from whole blood and treated with LPS, revealing an effect of hyperglycemia (time) without an effect of ketones (condition).

First, the authors should be commended on their transparency of data reporting - it is greatly appreciated. The manuscript in its present form is light on data and primarily reports null findings. Additional reporting of circulating biomarkers of inflammation to characterize the participants is requested. These analysis are cheap (standard chem-26 panel, CRP and cytokine multiplex) and easy to perform, and would add significant breadth to the data and importantly help interpret the ex vivo results. Additionally, speculations by the authors in the discussion could be adequately addressed with these assessments.

Additional comments and recommendations are provided below. I look forward to seeing a revised version of this manuscript. 

Introduction

Could be improved - in the current form, several items (e.g. PAMPS, DAMPS) are discussed that are not measured in the study or brought up again until the limitations section. This may be better placed deeper into the introduction or reserved for discussion. Ultimately, I did not get a good understanding of the relevance of NLRP3 activation in human health and disease or the importance of this research using a ketone monoester drink and an OGTT.

Methods

Why was BMI>29 included if the well established obesity cutoff is BMI>30?

Please provide a more detailed description of the exclusion criterion.

"following a low-carbohydrate diet"

This is a broad statement, as CHO intake <40% total kcals has been defined as "low-carbohydrate" in the literature. Is this excluding this dietary pattern or only very low carbohydrate ketogenic diets? How was diet assessed? i.e. was the assessed from the diet log or a different questionnaire?

" or intermittent fasting protocol"

Again, broad statement. Typical overnight fasting is ~10 hours. At what cutoff were individuals excluded and how was this assessed?

Were other special diets excluded? e.g. DASH diet, Mediterranean diet, etc How was this assessed?

 "following an intensive exercise regime"

Still, broad statement. Please define activity criterion. Were they sedentary by ACSM guidelines? Perhaps exercising less than X minutes of moderate-vigorous activity per week?

Given the female dominant data set and age-range, what was the menopausal status of the women? were any on hormone therapy?

24 hour food log

I appreciate the authors' effort to control diet between the visits. However, this is an odd food collection time frame. Standard of practice is to perform a 24-hour dietary recall by a registered dietitian or administer a 3-day dietary food log (i.e. food record). Please justify the use of the 24 hour food log in this setting and consider utilizing services of a registered dietitian for future trials.

Statistics

Why was homoscedasticity not assessed with a statistical test?

Results

What was the magnitude of hyperglycemia in these individuals? The obesity phenotype can be highly variable in response to an OGTT.

To improve the quality and quantity of data in the manuscript, please report standard circulating biomarkers of inflammation at -30, 0 and 60 minutes from the human KMD/OGTT experiment in table 1 (e.g. CRP, leukocyte counts, cytokines like TNFα and interleukins - a cytokine panel (multiplex) ELISA is recommended). This information is important to characterize the inflammatory status of the individuals and interpret the ex vivo results.

Discussion

This section could be bolstered by placing the results in context with additional literature. e.g. references are light. See recommendations below.

The ketone monoester drink dramatically increases blood ketones, even in the presence of hyperglycemia. This is not a normal physiologic condition. Please comment on the ketone concentrations achieved via the KDM vs dietary approaches followed by an OGTT, e.g. ketogenic diet (~1.5mM) or prolonged fasting. This will help bolster the language in the discussion surrounding the length of ketone exposure potentially impacting results.

Line 246, the authors state, "In the present study, we expected that individuals with obesity would be more permissive to detecting the potential effects of -OHB to inhibit NLRP3 activation." How was this to be assessed given no healthy control group was used for comparison?

Line 252, the authors state "Alternatively, the sample recruited for this study may have been relatively healthy individuals with obesity such that baseline levels of inflammation were too low to see an effect. " Is there any evidence to support this - for example fasting blood chemistry with CRP values for comparison with healthy ranges? It is otherwise difficult to assume this given that there is no healthy comparison group used in this study and normative values are not available for the ex vivo approach used.

Line 262, does not seem to agree with your data in figure 1, "ß-OHB levels were only ~2 mM" whereby the average shown in figure 1 is ~3 mM (2.96 to be precise).

Line 268, in vivo should be italicized.

Line 274 "TLR4-TRIF-RIPK1-FADD-CASP8-NLRP3 " is not defined nor appropriately discussed.

Lines 279-281, require references to verify these statements.

Figure 1.

The asterisk and number designations are offset in several images. Please improve alignment. It is difficult to see the mean in the images. Please modify the representation to be more apparent.

Figure 2.

It's a bit concerning that the -30 and 0 values are essentially the same physiologic state, yet significance was only observed between the -30 and 60 minute, but not 0 and 60 minute. Why might this be?

Supplemental Figures

Please supplement the discussion in the manuscript given the visually apparent difference in males vs females (I understand the study is not powered to assess this.), but the data are interesting and the manuscript is currently lacking depth. Adding discussion about potential sex-differences by comparing the delta values from -30 to 60 minutes in males vs females and discussing mechanisms that may underlie these sex-based differences would be of interest as supplemental data.

Reviewer 2 Report

The manuscript” The impact of acute ingestion of a ketone monoester drink on LPS-stimulated NLRP3 activation in humans with obesity” describes negative results of placebo-controlled, double-blind and randomized clinical trial. Obese individuals were supplemented with b-hydroxybutyrate to assess its potential effect on NLRP3 inflammasome activation in monocytes. To assure high quality of the paper several comments should be taken into consideration:

Major comments:

Clinical relevance of immunomodulatory properties of food-delivered compounds are controversial. Frequently, effects observed in in vitro studies and/or animal models are nor confirmed in the clinical trials. This issue should be properly discussed in the manuscript (see PMID: 31817726), as it is also observed in case of the submitted manuscript. Obtained negative data and study limitations are discussed. However, it would be beneficial to perform some experimental approaches to deliver evidence-based explanation of observed discrepancies. Stimulation of whole blood samples with LPS resulted in activation of several innate immune processes via TLR4, including induction of PGE2, which further can lead to inhibition of NLRP3 inflammasome activation in humans on a very individual basis (PMID: 25917098). It might be responsible for the big variations in the observed effects. Therefore, it would be beneficial to first isolate monocytes e.g. using CD14 beads and then perform inflammasome activation assay, assessing in the end activated (mature) caspase-1 and mature IL-1 beta. More detailed demographic table of study subjects should be provided. Heterogenous and small study group might be the main reason of negative results. Some of the participants were obese (BMI more than 35). Statistical power analysis should be calculated. Additional experimental and data analysis approaches to adjust for possible confounders or to normalize the data (e.g. to BMI?) should be considered. To assess baseline and glucose-induced inflammasome-related phenotype of the study groups it would be beneficial to measure IL-1band IL-18 levels in the plasma of the patients. To fully state no effects of b-OHB on inflammasome activation IL-1bproduction should be confirmed by mature IL-1 beta and mature caspase 1 by WB. Diet diversity is a significant confounder in this study. According to the methods section dietary questionnaires have been collected. It would be beneficial to include this information in the manuscript and account for those differences. Is it possible to calculate daily lipids/sugars intake? The methodology of assessing MFI of activated caspase-1 should be described in more detail. Isotype control and raw data histograms should be provided.

Minor comments:

Figure 1. Dots stating statistical significance are moved and should be aligned. Figure 3. Letters indicating figure parts, as well as dots stating statistical significance are moved and should be aligned. Supplementary figures 1 and 2. The sample number and colors between male and female plotted on the figures appeared to be mixed and should be corrected.

Round 2

Reviewer 2 Report

The manuscript” The impact of acute ingestion of a ketone monoester drink on LPS-stimulated NLRP3 activation in humans with obesity” describes negative results of a placebo-controlled, double-blind and randomized clinical trial. Obese individuals were supplemented with b-hydroxybutyrate to assess its potential effect on NLRP3 inflammasome activation. The manuscript improved its quality; however, several comments should be taken into consideration:

Major:

  1. Line 45. NLRP3 is not the only Pattern Recognition Receptor, that can recruit ASC protein and activate the inflammasome complex. It is not correct to refer to ASC as NLRP3-associated ASC-like protein. ASC protein should be referred to as “apoptosis-Associated speck-Like protein containing a CARD”. 
  2. Section 3.5. The section describes plasma cytokines in the study group. Table 1 does not include any data regarding plasma cytokines concentration. The section should refer to Table 2.
  3. The supplementary materials file is not complete. The reviewer could see only Supplementary figures 1-3. Supplementary Figures 4 and 5 are missing.
  4. For a better understanding of the obtained result, it would be beneficial to include Supplementary figures 1-3 in the body of the manuscript. 
  5. Limitations and future perspectives section should be strengthened with significant references (f.e. PMID: 31817726).

Minor:

  1. Line 266. Misspelling in the line should be corrected.